# Caspase-1-dependent spatiality in triple-negative breast cancer and response to immunotherapy

Weiyue Zheng [1,8], Wanda Marini[1,8], Kiichi Murakami[1], Valentin Sotov[1], Marcus Butler [1,2,3], Chiara Gorrini [1,4], Pamela S. Ohashi[1,5,6] & Michael Reedijk [1,6,7] ✉

Tumor immune microenvironment (TIME) spatial organization predicts outcome and therapy response in triple-negative breast cancer (TNBC). An immunosuppressive TIME containing elevated tumor-associated macrophages (TAM) and scarce CD8+ T cells is associated with poor outcome, but the regulatory mechanisms are poorly understood. Here we show that ETS1-driven caspase-1 expression, required for IL1β processing and TAM recruitment, is negatively regulated by estrogen receptors alpha (ERα) and a defining feature of TNBC. Elevated tumoral caspase-1 is associated with a distinct TIME characterized by increased pro-tumoral TAMs and CD8+ T cell exclusion from tumor nests. Mouse models prove the functional importance of ERα, ETS1, caspase-1 and IL1β in TIME conformation. Caspase-1 inhibition induces an immunoreactive TIME and reverses resistance to immune checkpoint blockade, identifying a therapeutically targetable mechanism that governs TNBC spatial organization.

Triple-negative breast cancer (TNBC), a clinical surrogate for basal-like breast cancer (BLBC), is an aggressive malignancy with poor prognosis and an unmet need for effective targeted treatment. TNBC, which lacks the expression of estrogen receptor (ER), progesterone receptor (PR), and HER2, accounts for only 20% of all breast cancers, yet is responsible for a relatively large proportion of breast cancer deaths. It primarily affects young women in the prime of life, people of African ancestry, and those with BRCA1 mutations.

Compared to less-aggressive breast cancer subtypes, TNBC are infiltrated by more immune cells, and the pattern of immune infiltration is strongly associated with outcome. High tumor-associated macrophage (TAM) count is inversely related to survival, while elevated cytotoxic T lymphocytes (CTL) are associated with improved survival[1]. In addition to predicting improved survival in TNBC, CTL number predicts increased response to radiotherapy, neoadjuvant and adjuvant chemotherapy[2]. These findings exposed an opportunity to treat these cancers through blockade of programmed cell death protein 1 (PD1) or cytotoxic T-lymphocyte-associated protein 4 (CTLA4), immune checkpoints that silence CTL response. That being said, immune checkpoint blockade (ICB) has only demonstrated modest activity, with objective response rates (ORR) in phase 1 studies for single-agent treatment of around 10%, and 40% when in combination with nab-paclitaxel[3,4]. In recent phase 3 clinical trials, ICB demonstrated an improvement in complete pathologic response (64.8% vs 51.2%) and overall response rate (56.0% vs 45.9%) when comparing standard chemotherapy plus ICB to chemotherapy alone, respectively[5,6]. These promising, yet less-than-satisfactory results support the continued pursuit of ICB-based technologies to treat these cancers. Immunosuppressive TAMs, which can suppress CTLs through ICB-independent

[1]Princess Margaret Cancer Centre, University Health Network, Toronto, ON, Canada. [2]Department of Medical Oncology and Hematology, Princess Margaret Cancer Centre, University Health Network, Toronto, ON, Canada. [3]Department of Medicine, Division of Medical Oncology, University of Toronto, Toronto, ON, Canada. [4]School of Molecular and Cellular Biology, University of Leeds, Leeds, UK. [5]Department of Immunology, University of Toronto, Toronto, ON, Canada. [6]Department of Medical Biophysics, University of Toronto, Toronto, ON, Canada. [7]Department of Surgical Oncology, Princess Margaret Cancer Centre, University Health Network, Toronto, ON, Canada. [8]These authors contributed equally: Weiyue Zheng, Wanda Marini. ✉e-mail: Michael.Reedijk@uhn.ca

mechanisms[7], are an obvious therapeutic target, whose elimination may reboot a sensitivity to ICB.

We have recently demonstrated that TNBC are uniquely capable of expressing IL1β due to aberrant activation of the Notch developmental signaling pathway, resulting in the recruitment of TAMs to the cancer, a reduction in activated CTLs, and cancer progression[8,9]. IL1β secretion requires two steps. First, "priming" involves the induction of mRNA and protein production of an inactive IL1β pro-protein (pro-IL1β). Classically this occurs in cells of the innate immune system in response to molecular motifs called "pathogen associated molecular patterns" (PAMPs) that are carried by invading microbes, however, in TNBC this is driven by Notch. In the second "cleavage" step of IL1β production, a multiprotein cytosolic "inflammasome" complex is assembled in response to PAMPs or "danger associated molecular patterns" (DAMPs). With assembly of this complex, which contains nucleotide-binding oligomerization domain, leucine rich repeat, and pyrin domain containing receptor (NLRP) and apoptosis-associated speck-like (ASC) proteins, the key constituent pro-caspase-1 (p45) is recruited, and activated[10]. Activated caspase-1 (p20/p10) promotes proteolytic cleavage, maturation, and secretion of IL1β.

Here we report that, while inflammasomes have traditionally been described in immune cells, activated caspase-1 is expressed in TNBC cells allowing IL1β maturation, macrophage recruitment, and tumor progression. With machinery in place to both prime and cleave, TNBC are uniquely suited for IL1β production. Herein we show that caspase-1 blockade reduces TAM infiltration and enhances T lymphocyte infiltration and sensitivity to ICB.

## Results

### ETS1-driven caspase-1 expression is a hallmark of TNBC

TNBC cells are characterized by Notch-dependant IL1β expression and Notch-independent expression of caspase-1[8,9], both prerequisites for IL1β maturation and the recruitment of pro-tumoral TAMs to the tumor microenvironment. Here we established in a large survey of breast cancer cell lines[11] and primary tumors[12,13] that ER and caspase-1 expression are inversely related (Fig. 1a, b; Fig. S1a, b). Proving a causal relationship between ER and caspase-1, siRNA-mediated down-regulation of ERα in hormone receptor-positive breast cancer cell lines resulted in an increase of both caspase-1 mRNA and pro-caspase-1 protein (Fig. S1c). Also, pro-caspase-1 expression emerged with ERα loss in TMX2-28 breast cancer cells, an ER-independent derivative of MCF7 isolated after prolonged tamoxifen treatment[14] (Fig. S1d, e). Conversely, ERα overexpression resulted in pro-caspase-1 down-regulation in TNBC cells and a decrease in secreted IL1β and IL18 in conditioned media (Fig. S1f) and, confirming the presence of functional caspase-1, a decrease in the ability of the media to promote macrophage migration (Fig. S1g).

In ER-negative colon carcinoma cells the transcription factor ETS1 drives caspase-1 expression[15], whereas in neuroblastoma and ERα-positive breast cancer, ERα sequesters ETS1 as a co-activator (together with nuclear receptor coactivator p160) and promotes transcription of estrogen response genes[16,17]. Based on these observations, we hypothesized that the paucity of ERα in TNBC allows unrestricted ETS1-dependent caspase-1 expression. ETS1 was more abundant in the BLBC phenotype, correlated with and was required for caspase-1 expression, and ectopic expression of ERα converted TNBC cells to a caspase-1-deficient phenotype (Fig. S2a–e; Fig. 1c, d). These findings were conserved in a murine TNBC cell line derived from mammary tumors of K14-cre, BRCA1[fl/fl], and p53[fl/fl] (KBP) mice[18] (Fig. S3). Caspase-1 expression could be induced in hormone receptor-positive breast cancer cells by increasing the ratio of ETS1:ERα (Fig. 1e, f).

Chromatin immunoprecipitation (ChIP) assays showed that ETS1 binding to the caspase-1 promoter was inversely dependent on ERα (Fig. 1g–i). ERα knockdown (KD) in luminal breast cancer cells resulted in increased binding of ETS1 to the caspase-1 promoter, while ERα over-expression in TNBC displaced ETS1 binding.

These findings indicate that through ETS1, ERα regulates caspase-1, which is a defining feature of breast cancer subtype.

### Caspase-1 is associated with the spatial immunophenotype in TNBC

The association between tumor cell caspase-1 and the tumor immune phenotype was explored in 24 primary breast cancers (12 ER+ and 12 triple negative) using multiplex IHC. Tissues were stained for pan-cytokeratin (panCK; tumor epithelium), caspase-1, CD3 (lymphocytes), CD8 (CTL), CD68+ (monocytes/macrophages), and CD163+ (M2-like, protumoral TAMs) (Fig. 2a, Fig. S4a, b). Caspase-1 positive cells and immune infiltrates were quantified in whole tumor, in tumor epithelial nests (CKpos) and in tumor stroma (CKneg). Confirming our previous findings, epithelial caspase-1 and IL1β were elevated in TNBC compared to ER positive tumors (Fig. 2b and Fig. S4c). Compared to ER+ breast cancer, immune infiltrates were enriched in TNBC and were predominantly in the stromal compartment (Fig. 2c, d, Fig. S4d, e). Accordingly, a positive correlation was identified between caspase-1 mRNA and lymphocyte infiltration in the TCGA PanCancer Atlas dataset (Fig. S4f, g). Examining TNBC further, M1-like macrophages were more numerous in the stroma of tumors with low epithelial caspase-1 expression, whereas stromal M2-like macrophages were elevated in caspase-1 high tumors. (Fig. 2e; Fig. S4h–j). Stromal expression of caspase-1, expected from immune cell infiltrates, was not associated with macrophage subtype (Fig. S4k–m). These findings suggest that caspase-1 expressed in cancer cells coordinates a class switch from classically activated (M1-like) to alternatively activated (M2-like) TAMs. TNBCs were grouped according to the location and extent of CD8+ infiltrates as previously described[19,20], with the finding that elevated epithelial caspase-1 expression was associated with restriction of CD8+ T cells to the stromal compartment ("stroma restricted") (Fig. 2f–h). Macrophages impede CD8+ T cells from reaching tumor cells[21], and our findings suggest that through caspase-1, tumor cells control T cell location by coordinating TAM polarization to an immunosuppressive M2-like phenotype.

### Caspase-1 drives IL1β-mediated macrophage recruitment and breast cancer growth

IL1β is classically described as a product of immune cells. As such, mouse tumor models of therapeutic IL1β inhibition[22] have not considered the key role of tumoral IL1β. To prove that IL1β processing mediated by caspase-1 within TNBC cells contributes to a functionally and spatially distinct TIME, KBP cells were reengineered with modifications in the IL1β processing pathway according to our findings in human TNBC cells (Fig S5a–d). As expected, caspase-1 did not affect *IL1β* gene expression, and because of the absence of paracrine stromal-tumor cell interaction, KBP cells grown in monolayer produced minimal IL1β[9] (Fig S5e, f). However, KBP cells isolated from KBP tumors produced 100-fold more IL1β, and this was caspase-1 dependent.

Given that KBP cells faithfully reproduced our findings in human TNBC cells, genetically modified KBP cells were orthotopically injected into the mammary fat pads of FVB/NJ mice to create mammary tumors. Compared to wild type, KBP allografts that over-expressed ERα, or in which ETS1, caspase-1 or IL1β were knocked down, demonstrated reduced growth rate, size, and F4/80+ macrophage infiltration (Fig. S6a–e). Treatment of wild type KBP-bearing mice with the caspase-1 antagonist VX-765 (Belnacasan®) produced similar results (Fig. S6f). Demonstrating the ability of VX-765 to induce an immunoreactive TIME within human tumors, treatment of humanized mice bearing MDA-MB231 xenografts decreased CD14+ HLA-DR+ macrophages and increased activated granzyme B positive (GrB+) CD8+ T cells, with a trend ($p = 0.1$) towards decreased tumor size (Fig. S6g).

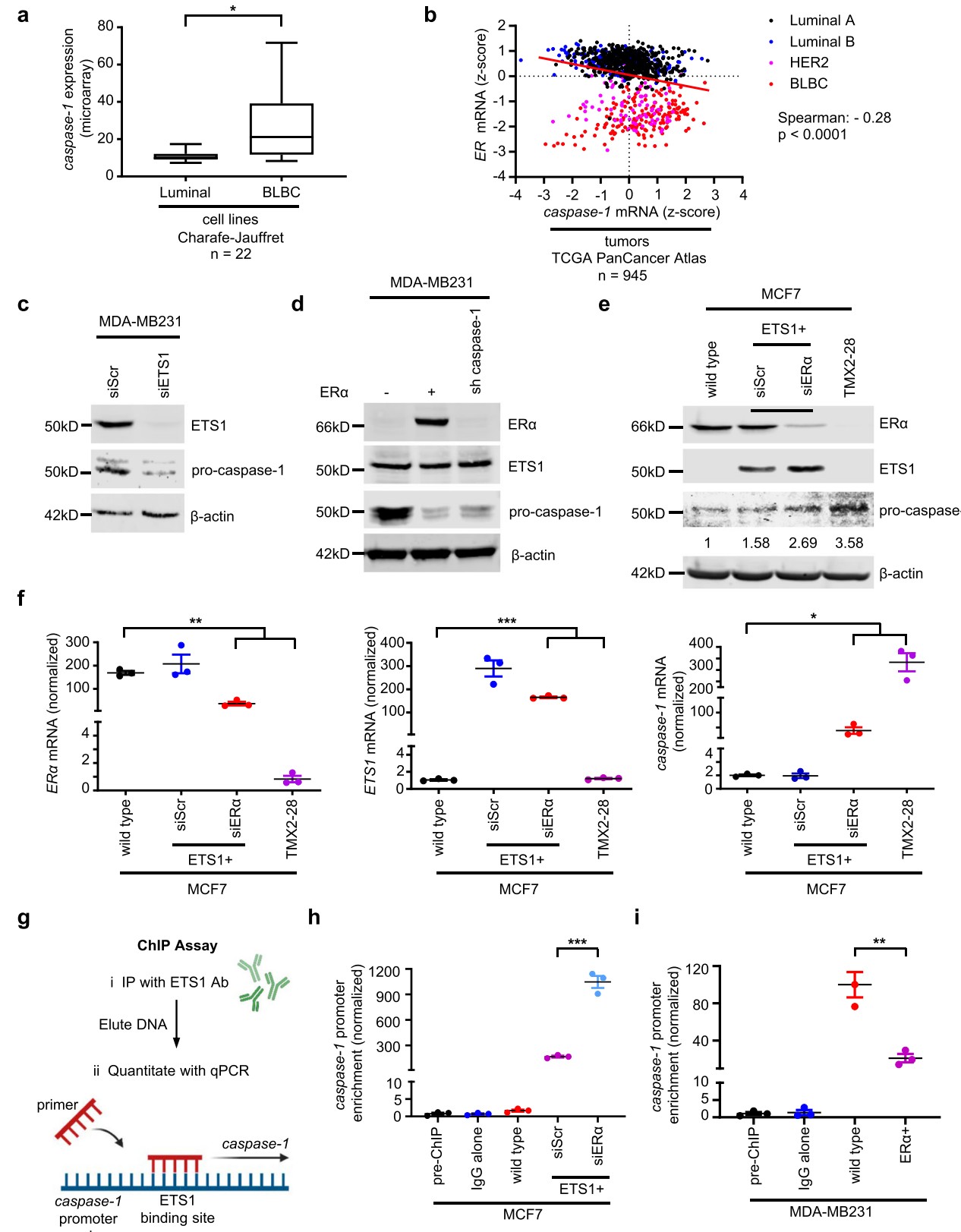

Since IL18[23] and gasdermin D[24] are also caspase-1 substrates, it was important to prove that IL1β was the key mediator of caspase-1 activity in our models. To this end, the effect of targeting caspase-1 was rescued in re-engineered KBP allografts where the caspase-1 cleavage site in the *IL1β* gene was replaced with P2A[25], allowing caspase-1-independent processing of IL1β (Fig. S7, Fig. 3a–e). The functional importance of caspase-1 in

CD8+ cell exclusion from the tumor core implied by our survey of human tumors, was confirmed using KBP allografts (Fig. 3f, g).

Building upon our previous work showing that Notch primes IL1β expression in TNBC, these data provide mechanistic insight into how IL1β is further processed in this breast cancer subtype, leading to TAM recruitment, CTL exclusion, and tumor progression.

**Fig. 1 | Caspase-1 expression is regulated by ETS1 and ERα. a** mRNA expression of caspase-1 in 22 (13 Luminal and 9 BLBC cell lines) human breast cancer cell lines[11] classified into two groups based on subtype (luminal vs BLBC), mean with box defining interquartile interval, 95% confidence interval (CI), two-tailed, unpaired t-test with Welch's correction for unequal variances; *: $p = 0.032$. **b** Association between caspase-1 and ER mRNA expression in 945 human breast cancer samples (TCGA PanCancer Atlas dataset[12,13] quantified by RNAseq, divided by subtype (Spearman correlation for all subtypes, two-sided). **c** Immunoblot of ETS1 and pro-caspase-1 from MDA-MB231 TNBC lysates after ETS1 knockdown (KD). **d** Immunoblot of ERα, ETS1 and pro-caspase-1 from MDA-MB231 cells without (-) or with (+) ectopic ERα expression or after sh-mediated caspase-1 KD. **e** Immunoblot of ERα, ETS1, and pro-caspase-1 in wild-type MCF7 cells, cells overexpressing ETS1 (ETS1+) without (siScr) and with ERα siRNA KD (siERα), and in TMX2-28 cells. β-

actin is included as a loading control. **f** RTqPCR analyses of ERα, ETS1, and caspase-1 in wild-type MCF7 cells, cells overexpressing ETS1 (ETS1+) without (siScr) and with ERα siRNA KD (siERα), and in TMX2-28 cells. One-way ANOVA with Dunnett's multiple comparisons test, *: $p = 0.0104$, **: $p = 0.0042$, ***: $p = 0.0003$. Data are presented as mean values ± standard error of the mean (SEM). **g** Schematic of ChIP assay for ETS1 binding to the *caspase-1* promoter. **h** ChIP assay from MCF7 cells after ectopic ETS1 expression and ERα KD, ***: $p = 0.0003$. **i** ChIP assay from MDA-MB231 cells after ectopic ERα expression, **: $p = 0.0053$. *Caspase-1* promoter/ETS1 complexes are quantified by RTqPCR after immunoprecipitation with nonspecific (IgG) or with anti-ETS1 antiserum. Mean with SEM, two-tailed, unpaired t-test with equal variances. $n = 3$ biologically independent experiments for immunoblots where representative images are shown, RTqPCR and ChIP assays.

## Caspase-1 blockade reduces TAM infiltration and enhances sensitivity to ICB

We hypothesized that by promoting TAM infiltration, caspase-1 induces an immune cold[26], ICB-refractory TIME. To address this therapeutic challenge, the KBP model was used to explore combined caspase-1 inhibition and ICB on the TIME and tumor progression. Wild type and caspase-1 KO KBP allograft-bearing mice were randomly allocated to treatment with either anti-PD1 ICB (RMP1-14) or control treatment (Fig. 4a–d; Fig. S8a). In contrast to caspase-1 KO, anti-PD1 treatment alone had no therapeutic benefit, supporting previous clinical studies showing a low response rate to ICB in TNBC. However, the addition of anti-PD1 further increased the effect of capase-1 KO on tumor growth, weight, and median survival. Capase-1 KO was associated with reduced F4/80+ macrophages, and reflecting our findings in human TNBC, a class switch from alternatively activated (CD206+, M2-like) to classically activated (CD206-, M1-like) TAMs. Additionally, combination therapy increased GrB+ CD8+ T cells. The findings were similar when caspase-1 was inhibited with VX-765 in KBP (Fig. 4e–h) and in a 4T1 murine TNBC allograft model (Fig. S8b–f).

To confirm the critical role of TAMs, KBP and KBP caspase-1 KO allografts underwent macrophage depletion by intraperitoneal injection of anti-CSF antibody[27] (Fig. 5a). KBP allograft growth in animals treated with anti-CSF-1 mimicked caspase-1 KO allograft growth, and anti-CSF-1 treatment was redundant in KBP caspase-1 KO allografts, identifying TAMs as the mediator of the caspase-1 effect (Fig. 5b–f). In these conditions the anti-tumor effect of combined anti-PD1 was maintained.

Overall, these findings suggest that by reducing TAM recruitment, caspase-1 inhibition switches the TIME to a hot, immunoreactive phenotype, establishing sensitivity to ICB.

## Discussion

Recent advances in spatial transcriptomics and imaging mass cytometry have revealed at high-resolution the phenotype, activation state, and spatial arrangement of cells in TNBC[28,29]. This has led to an improved understanding of how tumor architecture relates to outcome and response to therapy. An immunoreactive TIME characterized by CD8+ T cells, elevated expression of immune inhibitory molecules including PD1, PDL1 and Indoleamine 2, 3-dioxygenase, and accumulation of proinflammatory M1-like TAMs predicts good outcome[19]. The location of specific cells within the tumor and relative to each other provides additional prognostic and predictive information. A spatial phenotype that brings effector GrB+ CD8+ T cells in close proximity to malignant epithelium in the context of abundant immune inhibitory molecules (PD1–PDL1) and a type 1 interferon signature is associated with improved survival and response to ICB[19,20,29]. These cytotoxic interactions are lacking in immune cold and T cell stroma restricted TNBC phenotypes and likely explain reduced overall outcome, and the poor performance of ICB. A deeper understanding of the mechanisms that govern the configuration of the immune TIME is crucial to the development of improved precision immune-based

therapeutics. To that end we have shown that caspase-1-dependent IL1β production is a defining feature of TNBC that drives the recruitment of pro-tumoral TAMs, the exclusion of CD8+ T cells from the tumor core, tumor growth, and resistance to anti-PD1 immunotherapy. Indeed, stratifying TNBC according to caspase-1 expression shows that caspase-1-low tumors have a fully inflamed phenotype, characterized by an accumulation of M1-like TAMs and tumoral infiltration by CD8+ T cells. Confirming the mechanism, inhibition of caspase-1 signaling in mouse models of TNBC induces an immunoreactive TIME and reverses resistance to ICB. These findings illuminate the clinical potential of TAM-targeted therapies to promote a fully inflamed, ICB-sensitive TIME.

M2-like TAMs within the TIME play a pivotal role in tumorigenesis and metastasis and confer poor prognosis in TN/BLBC[1]. They promote cancer stemness, angiogenesis, tumor cell proliferation, migration, invasion, and immunosuppression, and are associated with resistance to chemotherapy and radiotherapy[30–33]. TAMs can suppress CD8+ T cells through both immune checkpoint-dependent and -independent mechanisms[7]. They can directly inhibit T cell responses through checkpoint engagement, production of inhibitory cytokines, and through depletion of metabolites, and production of reactive oxygen species. TAMs can also inhibit T cells indirectly by recruiting suppressive (Tregs) or by inhibiting stimulatory (dendritic cells) immune populations, or by regulating vascular or extracellular matrix barriers of T cell access to tumor cells. Our findings implicate TAMs as principal mediators of the caspase-1 effect on T cell location, state of activation, and response to ICB. The development of therapeutic strategies to target pro-tumoral TAMs is a necessary next step in the development of an effective immuno-armamentarium.

Caspase-1 exhibits pleotropic pro-tumoral activities and inhibiting it or its targets is a potential double-edged sword. Through its role in IL1β processing caspase-1 indirectly contributes to tumoral recruitment of TAMs and other pro-tumoral inflammatory cells[22], tumor growth, invasion, angiogenesis[34], metastases[35], stemness and epithelial-mesenchymal transition[36]. Contrary evidence suggests that IL1β has anti-tumor activities, specifically by inducing both Th1 and Th17 in myeloma and lymphoma[37]. Nevertheless, IL1β upregulation is generally associated with a poor prognosis in cancer[38]. Supporting IL1βs candidacy as an immunotherapeutic target, the phase 3 CANTOS trial, which tested the IL1β inhibitor canakinumab as a treatment for heart failure, demonstrated a greater than 50% reduction of death from all cancers[39]. In addition to IL1β, caspase-1 proteolytically cleaves the precursors of gasdermin D, IL18, and PPARγ. Gasdermin D induces pyroptosis, a mechanism of programmed cell death that activates strong inflammatory and immune responses and depending on context, has both pro-tumoral and anti-tumoral activities[40]. Although conflicting effects on cancer progression have also been described for IL18 and PPARγ, both can promote M2-like TAM differentiation, reversible by caspase-1 inhibition[41,42]. Additionally, IL18 increases the PD1-expressing immunosuppressive NK cell fraction and is associated with poor prognosis in TNBC[43]. Our findings in TNBC suggest that

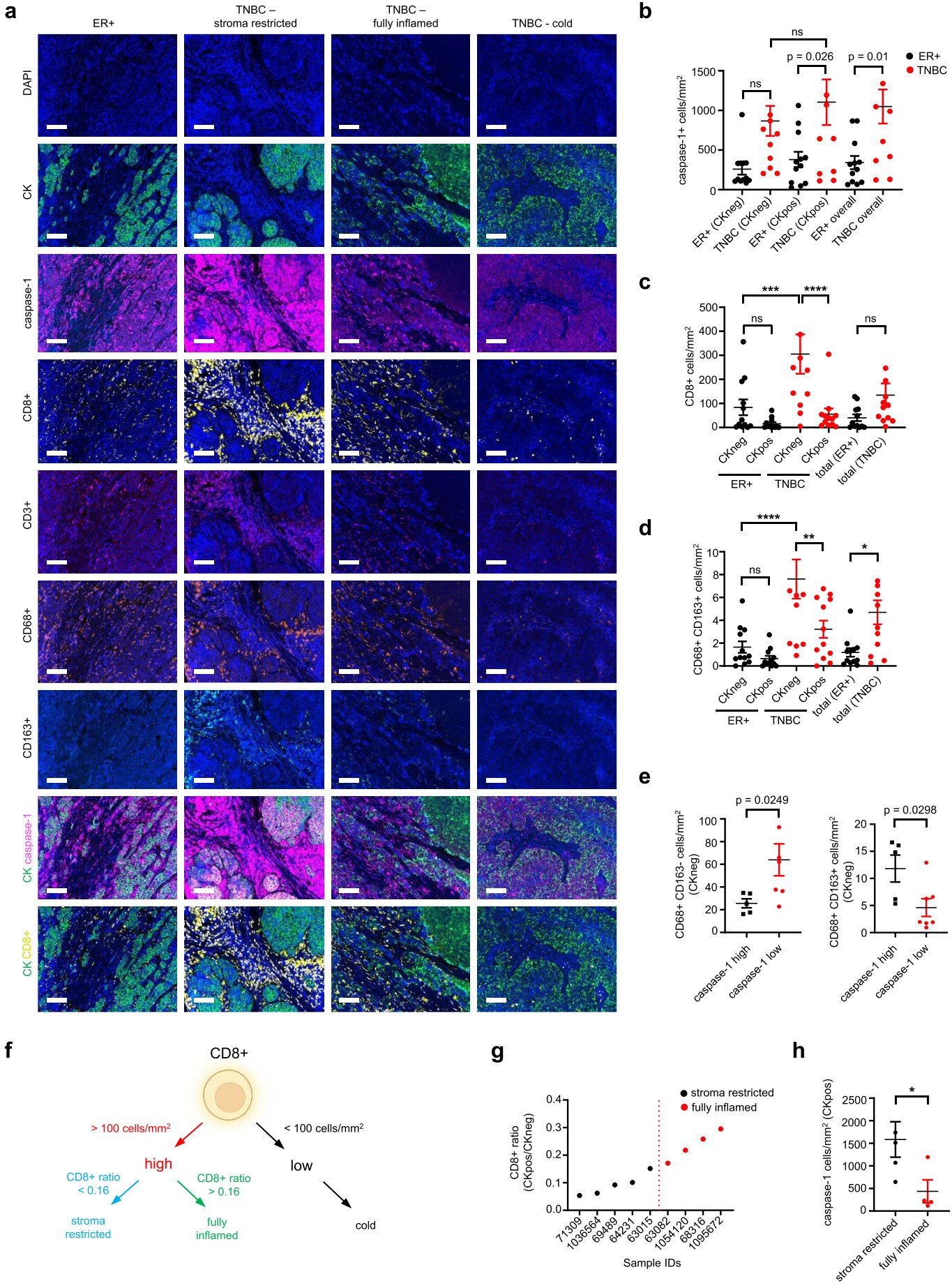

**Fig. 2 | Intratumoral caspase-1 is associated with spatial immunophenotype in TNBC. a** Representative images of multiplex IHC of ER positive samples, and TNBC samples according to their CD8+ T-cell spatial phenotype (DAPI: blue, CK: green, caspase-1: magenta, CD8+: yellow, CD3+: red, CD68+: orange, CD163+: teal; scalebar corresponds to 100 μm). **b** Multiplex IHC quantification of caspase-1 in the stromal compartment (CKneg), intratumoral compartment (CKpos), or both (overall) in ER positive (ER+) and TNBC tumor samples. Quantification of CD8+ T-cells, ***: $p = 0.0006$, ****: $p < 0.0001$ (**c**) and CD68+ CD163+ M2-like macrophages, *: $p = 0.01$; **: $p = 0.0054$. ****: $p < 0.0001$ (**d**) in ER+ and TNBC samples in the stromal (CKneg) and intratumoral compartments (CKpos), or overall (total) ($n = 12$ ER+,

$n = 12$ TNBC. **e** M1-like (CD68+ CD163-) and M2-like (CD68+ CD163+) macrophage quantification in the stroma of TNBC samples with high and low intratumoral caspase-1 ($n = 5$ caspase-1 high, $n = 7$ caspase-1 low). **f** Stratification of TNBC samples into spatial immunophenotype according to stromal CD8+ T-cell number (high vs low) and tumor/stroma CD8+ T-cell ratio. **g** Division of TNBC samples into stroma restricted ($n = 5$) and fully inflamed ($n = 4$) subgroups based on the CD8+ T-cell ratio median (red dashed line). **h** Intratumoral caspase-1 quantification in the stroma restricted ($n = 5$) and fully inflamed ($n = 4$) subgroups, *: $p = 0.0459$. Mean with SEM, one-way ANOVA with Bonferroni's multiple comparisons test for (**b**–**d**). Mean with SEM, two-tailed, unpaired t-test with equal variances for (**e**, **h**). ns not significant.

caspase-1 is a viable therapeutic target and a potential predictive biomarker in TNBC, but further confirmatory clinical trial work with an effective and specific inhibitor is necessary.

Several caspase inhibitors have been designed as therapeutic tools, but few have moved to clinical trial because of inadequate efficacy, lack of specificity, or toxicity[44]. Of the compounds targeting caspase-1, VX-740 showed good anti-inflammatory performance in phase 1 and phase 2 trials for rheumatoid arthritis but the trials were discontinued due to drug-related hepatotoxicity. VX-765 completed phase 2 trials for the treatment of psoriasis and epilepsy and while well-tolerated, did not shown sufficient efficacy for approval for clinical use. For effective clinical translation, improved drug design together with a deeper understanding of caspase-1 function in normal tissues is needed before this class of therapeutics can be considered for cancer treatment. In the meantime, alternative compounds such as canakinumab, or the highly selective recombinant IL1β antagonist anakinra[45] should be considered for combination therapy with ICB. These compounds could be tested within the setting of a clinical trial in patients with Stage II/III TNBC receiving standard-of-care pembrolizumab ICB/taxane-platinum/anthracycline-based chemotherapy. The addition of cytotoxic chemotherapy could also reduce tumor escape noted in our mouse experiments, where immune modulation alone, was undertaken.

Overall, we provide insights into the biology of TNBC and suggest that caspase-1 plays a pro-tumoral role and contributes to anti-PD1 resistance. TNBC are uniquely capable of IL1β production, with Notch providing IL1β priming and ETS1-dependent caspase-1 expression ensuring IL1β processing, making TNBC an ideal candidate for IL1β blockade. The findings illuminate opportunities for combination immunotherapy with ICB. If successful, this therapeutic approach could be used in other malignancies where macrophages are a key barrier to therapeutic efficacy.

## Methods
### Database analysis
Microarray gene expression data from human breast cancer cell lines was downloaded from ref. 11. Human breast cancer tumor microarray (METABRIC) and RNAseq (TCGA PanCancer Atlas) gene expression and outcome data was obtained from cBioPortal[12,13].

### Cell culture
T47D, MCF7, MDA-MB231, HEK293T, and THP-1 cell lines were purchased from the American Type Culture Collection (ATCC). The tamoxifen-resistant MCF7 cell lines TMX-3-2, 3-6, 3-11, and 2-28 were a kind gift from Dr. John Gierthy (University of Albany, Albany, NY, USA) and generated after continuous exposure to tamoxifen as previously described[14]. The mouse KBP cell line was a kind gift from Dr. Chiara Gorroni (Ontario Cancer Institute, University Health Network, Toronto, ON, Canada) and created as previously described[18]. All MCF7 cells were cultured in Modified Eagle's Medium (MEM) (Gibco), supplemented with 10% fetal bovine serum (FBS) (Wisent Bioproducts), and 1% penicillin/streptomycin (P/S) (Wisent Bioproducts). MDA-MB231 cells were cultured in Dulbecco's Modified Eagle's Medium (DMEM) (Gibco) with 10% FBS, 1% P/S

and 2mM L-Glutamine (Gibco). HEK293T cells were cultured in DMEM (Gibco), supplemented with 10% FBS and 1% P/S. THP-1 cells were cultured in RPMI 1640 media (Gibco), supplemented with 10% FBS, 1% P/S, and 55 μM β-mercaptoethanol (Gibco). KBP cells were cultured in DMEM/F12 (Gibco) with 10% FBS, 1% P/S, 5 ng/mL epidermal growth factor (EGF) (Wisent Bioproducts), 10μg/mL insulin (Wisent Bioproducts), and 500μg/mL hydrocortisone (Sigma). All cells were maintained in 37 °C in 5% $CO_2$. To disassociate adherent cells, cells were washed with 1x phosphate buffered saline (PBS) (Gibco), and trypsinized with 0.05% trypsin/0.53 mM EDTA (Wisent Bioproducts) for 5 min. Cell lines were never passaged for greater than 6 months.

### Transient transfection
For siRNA experiments, cells were seeded at $5 \times 10^5$ cells/well in a 6-well plate and treated with 40 nM siRNAs (ON-TARGETplus siRNA SMARTpool, Dharmacon) using Lipofectamine RNAiMAX (Invitrogen) according to the manufacturer's protocol. Sequences of siRNAs are listed in Table S1.

For transient transfection with an ERα-expressing plasmid, cells were seeded at $5 \times 10^5$ cells/well in a 6-well plate and transfected with various concentrations of the mouse ERα plasmid on a pcDNA3.1 backbone (Genscript, catalog #OMu23198D) and empty pcDNA3.1 vector (ThermoFisher, catalog #V79020) using the X-tremeGENE HP DNA transfection agent (Roche) according to the manufacturer's protocol.

### Real-time quantitative polymerase chain reaction
RNA from cells was isolated using TRI Reagent (Sigma), Direct-zol RNA prewash and RNA wash buffers (Zymo Research), Econospin collection tubes (Epoch Lifescience), and eluted in RNAase-free water (Gibco). Reverse transcription of total RNA was carried out using the SuperScript VILO MasterMix (Invitrogen) with 1μg of RNA. Real-time quantitative polymerase chain reaction (RTqPCR) was performed with SYBR Green Advanced qPCR Master Mix (Wisent Bioproducts) as per the manufacturer's instructions, using the 7900HT Fast real-time PCR system (Applied Biosystems). Dissociation curve analysis was also performed to ensure the absence of non-specific amplification. Normalization was done using β-actin expression levels. All primers were ordered from Integrated DNA Technologies (IDT) (see Table S1 for details).

### Immunoblotting
Cells were lysed in RIPA buffer (25 mM Tris pH 7.6, 150 mM NaCl, 1% NP-40, 1% DOC, 0.1% SDS), sonicated and centrifuged at 1400 rpm for 10 min. Protein was quantified using the DC Protein Assay (Bio-Rad) on a Biotek Elx800 Microplate Reader according to the manufacturer's instructions. Proteins were resolved by SDS–polyacrylamide gel electrophoresis and blotted on a nitrocellulose membrane (Bio-Rad). Blots were incubated overnight in the corresponding primary antibodies followed by detection with fluorescent secondary antibodies (antibody details listed in Table S2). Proteins of interest were visualized on the Li-cor Odyssey imaging system and analyzed using the Image studio Lite software.

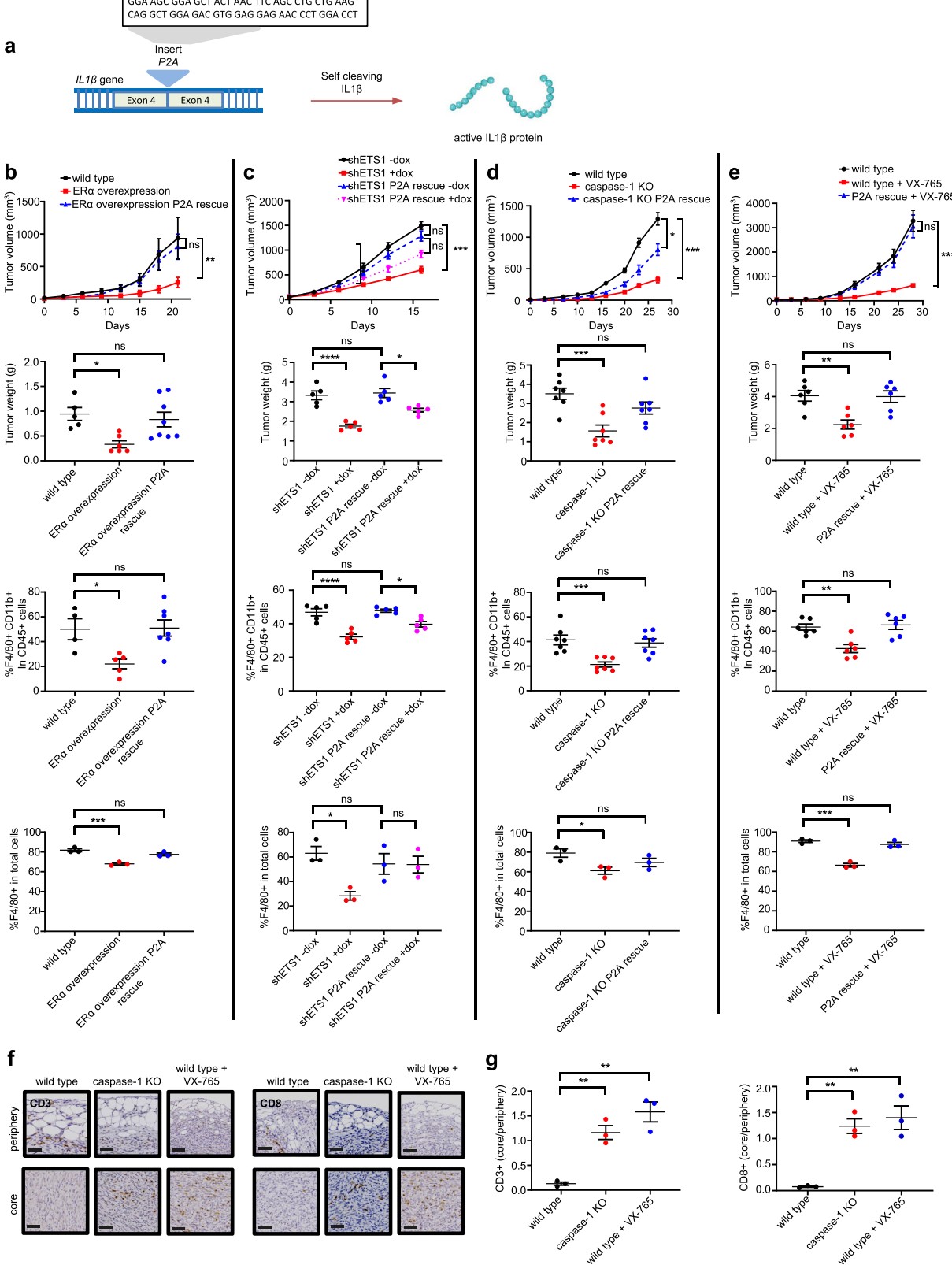

## IL1β immunoprecipitation

An IL1β antibody (R&D Systems) was first bound to Dynabeads Protein G (Invitrogen). Cells were lysed in cell lysis buffer (50 mM Tris-HCl pH 7.4, 10 mM NaCl, 5 mM EDTA pH 7.4, 0.4% NP-40, 5 mM NaF, 10% glycerol) and were added to the antibody-Protein G complex and incubated overnight at 4 °C. Elution of the target protein was done using 2x protein loading buffer (240 mM Tris-HCl pH 6.8, 6% SDS, 30% glycerol, 16% β-mercaptoethanol, 0.06% bromophenol blue) at 95 °C for 3 min, and analyzed using immunoblot.

**Fig. 3 | Caspase-1-mediated IL1β processing within TNBC cells contributes to a functionally and spatially distinct TIME. a** Schematic of IL1β-P2A where P2A[25] is inserted at the caspase-1 cleavage site within the *IL1β* gene, allowing caspase-independent processing of IL1β via endogenous ribosomal skipping. Created with BioRender.com, released under a Creative Commons Attribution-NonCommercial-NoDerivs 4.0 International license. Growth curves, tumor weights, flow cytometric quantification (F4/80 and CD11b double positive cells as a proportion of CD45 positive immune cells) and IHC quantification of F4/80 positive cells in wild type ($n = 5$), ERα-overexpressing ($n = 6$) and ERα-overexpressing P2A rescue KBP ($n = 8$) allografts, tumor weight *: $p = 0.024$, F4/80+ CD11b+ in CD45+ *: $p = 0.0137$, **: $p = 0.0042$, ***: $p = 0.00027$ (**b**), in KBP shETS1 allografts with and without P2A rescue in mice fed regular (-dox) or doxycycline-containing (+dox) diet at tumor onset ($n = 5$ mice/group), tumor weight *: $p = 0.0446$, F4/80+ CD11b+ in CD45+ *: $p = 0.0143$, F4/80 in total cells *: $p = 0.0197$, ***: $p = 0.0007$, ****: $p < 0.0001$ (**c**), in wild type, caspase-1 KO and caspase-1 KO P2A KBP allografts ($n = 7$ mice/group), tumor volume *: $p = 0.0326$, F4/80 in total cells *: $p = 0.0316$, tumor volume ***: $p = 0.0001$; tumor weight ***: $p = 0.0001$; F4/80+ CD11b+ in CD45+ ***: $p = 0.0007$ (**d**), and in mock-treated wild type KBP, VX-765-treated KBP and VX-765-treated P2A rescue KBP allografts ($n = 6$ mice/group), tumor weight **: $p = 0.0039$; F4/80+ CD11b+ in CD45+ **: $p = 0.004$, ***: $p = 0.0002$, ****: $p < 0.0001$ (**e**). Representative IHC images (**f**) and quantification (**g**) of the location (expressed as percentage of total cells in tumor core/tumor periphery) of CD3+ and CD8+ cells in wild type, caspase-1 KO and VX-765-treated KBP allografts ($n = 3$ mice/group), CD3+ wild type vs caspase-1 KO: **: $p = 0.0067$; wild type vs wild type + VX-765: **: $p = 0.0011$; CD8+ wild type vs caspase-1 KO: **: $p = 0.0053$; wild type vs wild type + VX-765: **: $p = 0.0027$; Mean with SEM, two-way ANOVA with Tukey's multiple comparisons test for growth curves (**b**–**e**), one-way ANOVA with Dunnett's multiple comparisons test for tumor weights and immune cell quantification (**b**, **d**, **e**, **g**), one-way ANOVA with Tukey's multiple comparisons test for tumor weights and macrophage quantification (**c**). ns = not significant, scalebar corresponds to 50µm.

## Cell line creation using lentivirus

Lentivirus was produced in HEK293T cells by co-transfecting packing plasmids pCMVdelta8.9 (Addgene) and pCMV-VSV-G (Addgene, catalog #8454), and the plasmids listed below using X-tremeGENE HP DNA transfection agent (Roche). After 48–72 h of co-transfection, supernatant was collected and purified through a 0.45 µm filter. Harvested virus was then added to cell of interest for 24 h, after which the appropriate antibiotic selection was carried out. Appropriate integration of plasmid was confirmed with immunoblot.

**MDA-MB231 ERα overexpression.** The human ERα plasmid was purchased from Addgene (catalog #28230) and cloned into the pBABE lentiviral vector (Addgene, catalog #1764) using the BamHI and EcoRI restriction enzymes (New England Biolabs).

**MCF7 ETS1 overexpression.** The human ETS1 plasmid was purchased from Addgene (catalog #82118) and cloned into the pBABE lentiviral vector (Addgene, catalog #1764) using the BamHI and EcoRI restriction enzymes.

**MDA-MB231 sh caspase-1.** The human shRNA lentiviral vector for caspase-1 was purchased from Addgene (catalog #53575).

**KBP ERα overexpression.** The mouse ERα plasmid was purchased from Genscript (catalog # OMu23198D) and cloned into the pHIV-Zsgreen lentiviral vector (Addgene, catalog #18121) using the BamHI and EcoRI restriction enzymes.

**KBP doxycycline inducible shETS1.** The mouse inducible shETS1 plasmid was purchased from Dharmacon (SMARTvector inducible lentiviral shRNA, catalog # V3SM7671-231267773).

All primers used for cloning and shRNA sequences are listed in Table S1.

## Cell line creation using CRISPR

**KBP caspase-1 KO.** The caspase-1 single guide RNA (sgRNA) sequence (IDT) was cloned into the PX458 Cas9-containing plasmid (Addgene, catalog #48138) using the XbaI and KpnI restriction enzymes (New England Biolabs). KBP cells were seeded at $5 \times 10^5$ cells/well in a 6-well plate and transfected with the caspase-1 sgRNA containing plasmid using the X-tremeGENE HP DNA transfection agent (Roche) according to the manufacturer's protocol.

**KBP IL1β KO.** The Cas9 protein (spCas9 2NLS nuclease, Synthego) and IL1β sgRNA (Synthego, catalog #311491) were transfected into KBP cells using Lipofectamine CRISPRMAX (ThermoFisher) according to the manufacturer's protocol.

The cells were sorted to clonality, and successful KO was confirmed with immunoblot and sequencing. All primers and sgRNA sequences are listed in Table S1.

## P2A rescue cell lines

**KBP P2A rescue.** The Cas9 protein (spCas9 2NLS nuclease, Synthego), IL1β sgRNA (Synthego, catalog #311491), and single stranded oligodeoxynucleotide (ssODN) containing the P2A sequence and homologous arm (IDT) were transfected into KBP cells using Lipofectamine CRISPRMAX (ThermoFisher) according to the manufacturer's protocol.

**KBP caspase-1 KO P2A rescue.** The Cas9 protein (spCas9 2NLS nuclease, Synthego) and caspase-1 sgRNA (Synthego, catalog #5299289) were transfected into KBP P2A rescue cells using Lipofectamine CRISPRMAX (ThermoFisher) according to the manufacturer's protocol.

**KBP ERα overexpression P2A rescue.** KBP P2A rescue cells were integrated with the mouse ERα plasmid using lentiviral transduction (described above).

**KBP doxycycline inducible shETS1 P2A rescue.** KBP P2A rescue cells were integrated with the mouse inducible shETS1 plasmid using lentiviral transduction (described above).

Integration of the P2A sequence was confirmed with sequencing. Successful KO was confirmed with immunoblot and sequencing, and overexpression or knock down was confirmed with immunoblot. All sgRNAs and the ssODN sequence are listed in Table S1.

## Chromatin immunoprecipitation

The ChIP assay was performed using the Imprint Chromatin Immunoprecipitation kit (Sigma-Aldrich) according to the manufacturer's instructions. In brief, cells were fixed and crosslinked by formaldehyde, and nuclear DNA was broken into small fragments by sonication. A ChIP grade ETS1 antibody (Cell Signaling) was used for immunoprecipitation of ETS1, following the elution of ETS1-bound DNA. The amount of caspase-1 DNA was quantitated using RTqPCR using primers for the caspase-1 promoter (IDT). A non-specific IgG (Cell Signaling) was used as a control. All primers and antibodies are listed in Tables S1 and S2.

## Transwell migration assay

The transwell migration assay was carried out using the QCM 24-well fluorimetric chemotaxis cell migration assay kit (EMD Millipore). Briefly, THP-1 cells were seeded on a basement membrane matrix-coated transwell insert (8 µm pore size) at a concentration of

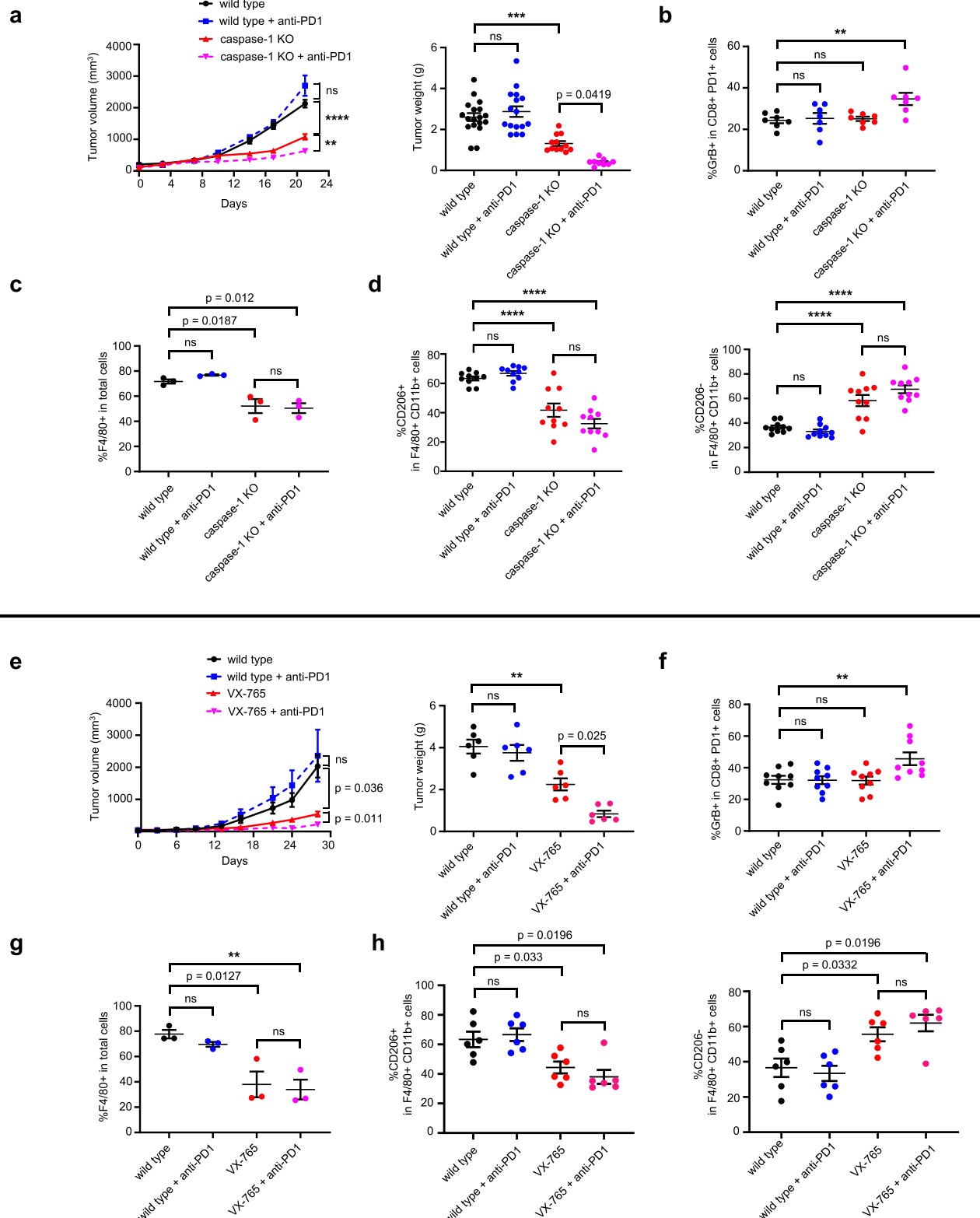

5 × 10⁴ cells/well. Phorbol 12-myristate 13-acetate (PMA) (Sigma) at 100 ng/mL was added to the culture media to induce conversion of monocytes to macrophages. After 24 h, PMA-containing media was replaced with regular media and after an additional 48 h, the media in the bottom chamber was changed to a conditioned media: control, media with 10 ng/mL recombinant IL1β protein (Acrobiosystems, clone ILB-H4110), media with 100 ng/mL recombinant IL1β, conditioned media from MDA-MB231 cells, conditioned media from MDA-MB231

ERα overexpressing cells, and conditioned media from MDA-MB231 sh caspase-1 cells. After 24 h in the presence of the conditioned media, THP-1 cells in the bottom chamber were collected. The number of cells were quantified as per the manufacturer's instructions.

**ELISA**

To determine secreted IL1β levels in cell culture supernatant, conditioned media was collected and centrifuged at 500 × *g* for 10 min at 4 °C, after

**Fig. 4 | Caspase-1 KO in KBP tumors improves the effect of anti-PD1 treatment.** **a** Growth curves and tumor weights of wild type and caspase-1 KO KBP orthotopic tumors treated with IgG control ($n = 18$ wild type, 12 caspase-1 KO) or anti-PD1 antibody ($n = 16$ wild type, 9 caspase-1 KO), **: $p = 0.0018$; ***: $p = 0.0002$, ****: $p < 0.0001$. **b** Flow cytometric quantification of activated CD8+ T-cells (GrB+ as a proportion of CD8 + PD1+ cells, **: $p = 0.0041$, $n = 7$. **c** Macrophage infiltration quantified by IHC (F4/80 as a proportion of total cells, $n = 3$. **d** CD206+ and CD206- cells as a proportion of F4/80+/CD11b+ cells, ****: $p < 0.0001$, $n = 10$. **e** Growth curves and tumor weights of wild type and VX-765 treated KBP orthotopic tumors treated with IgG control or anti-PD1 antibody ($n = 6$ mice/group), **: $p = 0.0023$. **f** Flow cytometric quantification of activated CD8+ T-cells (GrB+ as a proportion of CD8+ PD1+ cells, **: $p = 0.0053$, $n = 9$). **g** Macrophage infiltration quantified by IHC (F4/80 as a proportion of total cells, **: $p = 0.0073$, $n = 3$). **h** CD206+ and CD206- cells as a proportion of F4/80+/CD11b+ cells, $n = 6$. Mean with SEM, two-way ANOVA with Tukey's multiple comparisons test for growth curves, one-way ANOVA with Tukey's multiple comparisons test for tumor weights and macrophage quantification, one-way ANOVA with Dunnett's multiple comparison test for flow cytometry. ns not significant.

which the supernatant was isolated and concentrated using Amicon Ultra-4 centrifugal filters (EMD Millipore), and analyzed using the Quantikine ELISA kit (R&D systems). To determine secreted IL1β levels in mouse tumors, tumor cells were first isolated from immune and other non-malignant cells by flow cytometry, plated on a 6-well culture dish, and incubated overnight. The supernatant was collected and secreted IL1β was analyzed using the same ELISA kit described above.

### Humanized mouse line
Female Hu-NSG™ (JAX:005557) mice engrafted with human CD34+ at 16–18 weeks of age were purchased from Jackson Laboratory.

### Orthotopic tumor injections
Six-week old female FBV, BALB/c, and 16–18 weeks old CD34+Hu-NSG™ mice were purchased from The Jackson Laboratory and maintained in a pathogen-free environment at the Ontario Cancer Institute, University Health Network. All mouse protocols were approved by the Animal Care and Use Committee of the University Health Network. Prior to orthotopic injection, mice were anesthetized using inhaled 2.5% isoflurane. A small midline preperitoneal incision was performed to expose the mouse mammary fat pads. Cells to be used (KBP cells used for FBV mice; 4T1 cells used for BALB/c; and MDA-MB231 cells used for CD34+Hu-NSG™) for orthotopic injection were washed with $1 \times$ PBS, dissociated with trypsin, and resuspended in Cultrex Basement Membrane Extract (BME) type 3 (R&D Systems) at a concentration of $1 \times 10^6$ cells in 20μL of BME (KBP cells), $2.5 \times 10^5$ cells in 15 μL PBS (4T1 cells), or $1 \times 10^7$ cells in 15μL PBS (MDA-MB231 cells). The prepared cells were injected directly into the exposed mammary fat pad using a Hamilton syringe (Hamilton). The incision was closed with staples and mice were given post-procedure Amoxicillin water for 2 weeks to prevent a surgical site infection. Mouse tumors were measured several times a week using caliper measurements. Tumor volume was determined using the following formula: volume = (length × width²)/2, where the width represented the smaller diameter. Mice were euthanized and tumors were excised to target a humane endpoint of 1.5 cm in diameter according to ethical mouse protocols approved by the University Health Network Animal Care and Use Committee (ACUC; Toronto). In some cases, with permission of, and close monitoring by the ACUC, the target endpoint was exceeded. Ready to use humanized CD34+Hu-NSG™ mice were purchased from Jackson laboratory (strain: 705557). Use of this mouse line was approved by the Animal Care and Use Committee of the University Health Network.

### In vivo drug treatments
Once orthotopic tumors were palpable, mice were randomized to treatment groups. Doxycycline (625 mg/kg) was administered through the rodent diet (Teklad) and changed twice a week. VX-765 (MedChemExpress, catalog #HY-13205) was resuspended in 10% DMSO/90% corn oil (Sigma) and injected intraperitoneal (i.p.) at a concentration of 25 mg/kg every 3 days. 10% DMSO/90% corn oil was used as a vehicle control for VX-765 experiments. Mouse anti-PD1 monoclonal antibody (0.2 mg/mouse, BioXCell, clone RMP1-14) was injected i.p. every 3 days as previously described[46,47]. Mouse anti-CSF1 monoclonal antibody (BioXCell, clone 5A1) was injected at a loading dose of 1.0 mg/mouse i.p. followed by a maintenance dose of 0.5 mg/mouse i.p. every 5 days

as previously described[48,49]. A similar concentration of rat IgG2a isotype antibody (BioXCell, catalog #BE0089) was used as a control for both anti-PD1 and anti-CSF1 antibody experiments.

### Caspase-1 activity assay
The Caspase-Glo® 1 Inflammasome Assay (Promega) was used to measure caspase-1 activity. Briefly, cells were plated in a 96-well plate at a concentration of $6 \times 10^4$ cells/well. The Caspase-Glo® 1 Reagent was added according to the manufacturer's instructions, and the luminescence signal was read after 1 h of incubation time on the Hidex Sense Plate Reader using the Hidex Sense Plate Reader Software V1.3.0.

### Flow cytometry
Tumors were minced and incubated in digestion buffer [1 mg/mL of collagenase (Sigma) and 10 μg/mL of Pulmozyme (Roche), 2 mM L-glutamine (Lonza), 100 μg/mL P/S (Lonza) in Iscove's Modified Dulbecco's Medium (IMDM), Gibco] at 37 °C for 40–60 min. The digested samples were filtered through a 70μm Falcon cell strainer, stained with a fixable viability dye and fluorophore-conjugated antibodies (listed in Table S2). Immune infiltrates were analyzed with a BD Biosciences LSR Fortessa Analyzer using FlowJo software (TreeStar).

### Immunohistochemistry
Mouse mammary tumors were excised and fixed in 10% formalin for 24 h. Tumors were then embedded in paraffin and cut into 4 μm tissue sections. The sections were dewaxed and heat activated antigen retrieval was performed using 10 mM pH 6.0 citrate. Sections were incubated with a mouse F4/80 antibody (ThermoFisher) overnight at 4 °C, and a secondary antibody (VECTASTAIN ABC-HRP kit, Rat IgG Peroxidase, Vector Laboratories) for 45 min. Sections were developed with the DAB kit (Vector laboratories) and counterstained with hematoxylin. Stained slides were scanned at 40× on a whole slide scanner (Nanozoomer 2.0-HT, Hamamatsu, Japan) to acquire whole slide images. Quantification of IHC whole slide images was performed using the QuPath software.

### Multiplex immunohistochemistry
Formalin fixed paraffin-embedded (FFPE) human mammary tumor sections (Table S3) were obtained from the University Health Network (UHN) BioBank and analyzed by a pathologist for adequate quality. Slides were stained with the Opal 7-color automation IHC kit (Akoya Biosciences) according to the manufacturer's instructions. The following antibodies were used: panCK (clone AE1/AE3, Agilent), caspase-1 (EMD Millipore), CD3 (clone 2GV6, Roche), CD8 (clone SP57, Roche), CD68 (clone KP1, Agilent), and CD163 (clone 10D6, Biocare Medical). Full antibody details are listed in Table S2. Slides were annotated and scanned at high resolution using the Akoya Vectra 3.0 microscope. Machine learning-based automated image analysis was performed using the inForm Tissue Analysis Software (Akoya Biosciences). Corresponding slides were stained using hematoxylin and eosin (H&E) staining by the histology facility. Breast cancer subtype was acquired from the corresponding UHN Biobank IHC staining for ER, PR, and HER2, and corresponding patient data was extracted from the UHN Cancer Registry. Human ethics approval

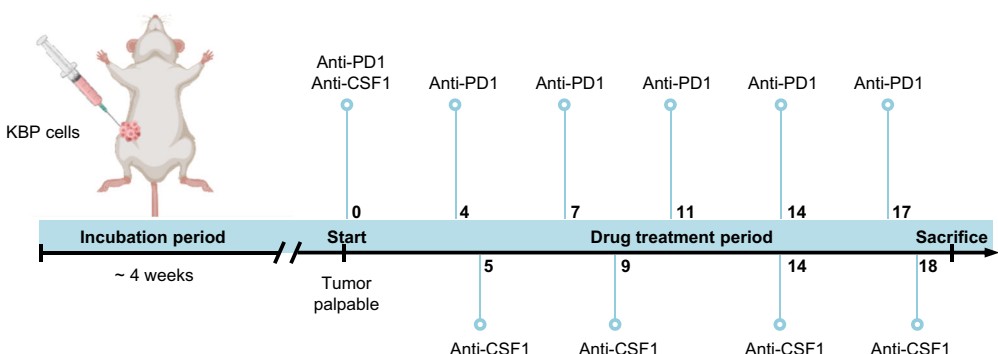

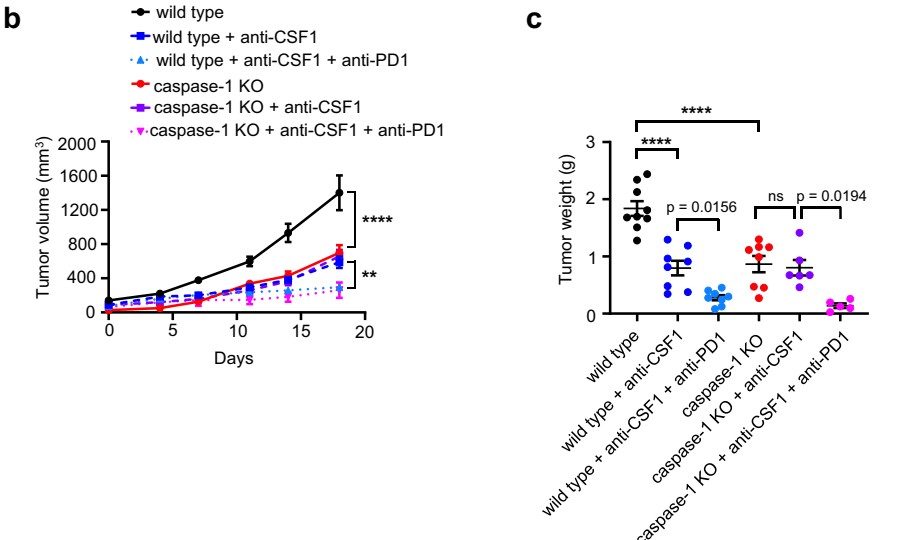

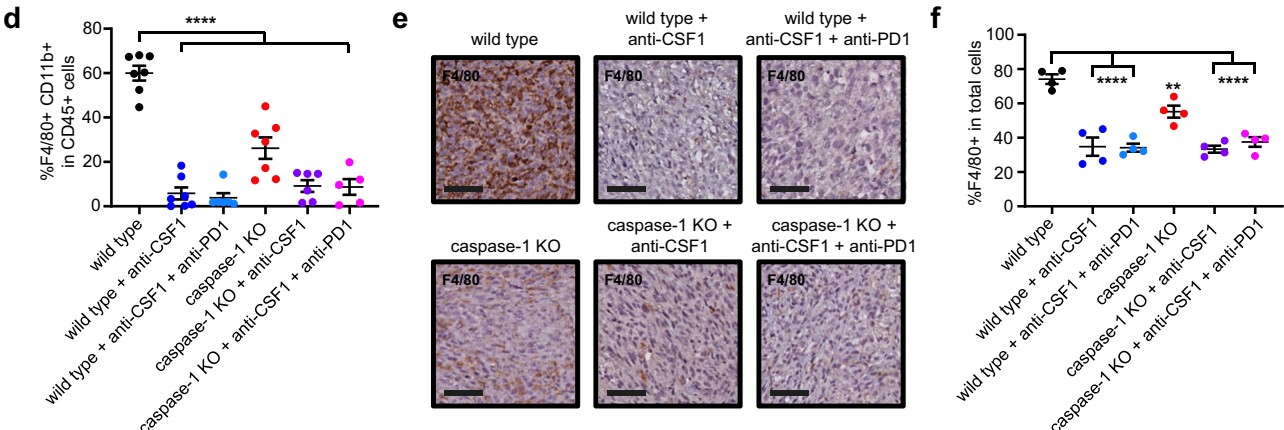

**Fig. 5 | TAMs mediate the effect of caspase-1 in ICB resistance. a** Schematic demonstrating the experimental design: KBP wild type or caspase-1 KO cells were injected into the mammary fat pads of syngeneic mice, with anti-PD1 or anti-CSF1 antibody treatment commencing once the tumors were palpable. Numbers represent drug injection days. Created with BioRender.com, released under a Creative Commons Attribution-NonCommercial-NoDerivs 4.0 International license. Growth curves, **: $p = 0.0019$; ****: $p < 0.0001$ (**b**) and tumor weights, ****: $p < 0.0001$ (**c**) of wild type and caspase-1 KO KBP tumors treated with IgG control ($n = 9$ wild type, 8 caspase-1 KO), anti-CSF1 antibody ($n = 8$ wild type, 6 caspase-1 KO), or combination anti-CSF1 and anti-PD1 antibody ($n = 8$ wild type, 5 caspase-1 KO). Flow cytometric quantification of macrophages (F4/80 and CD11b double positive cells as a proportion of CD45 positive immune cells, ****: $p < 0.0001$, $n = 7$) (**d**) and IHC staining (**e**) and quantification, (**: $p = 0.0084$; ****: $p < 0.0001$, $n = 4$) (**f**) of macrophage infiltration. Mean with SEM, two-way ANOVA with Tukey's multiple comparisons test for growth curves, one-way ANOVA with Tukey's multiple comparisons test for tumor weights, and one-way ANOVA with Dunnett's multiple comparisons test for macrophage quantification. ns not significant, scalebar corresponds to 50 µm.

was obtained from the University Health Network Research Ethics Board (Toronto).

For IL1β and pan-CK double staining, human mammary tumor sections were dewaxed and heat activated antigen retrieval was performed using 10 mM pH 6.0 citrate. Sections were incubated with a human IL1β antibody (Abcam) and pan-CK antibody (Abcam) overnight at 4 °C, and a two different secondary antibodies (VECTASTAIN ABC-HRP kit, Mouse IgG Peroxidase, Vector Laboratories; VECTASTAIN ABC-AP kit, Rabbit IgG alkaline phosphatase, Vector Laboratories) for 45 min. Sections were developed with the DAB kit and AP kit (Vector laboratories) sequentially. Stained slides were scanned at 40× on a whole slide scanner (Nanozoomer 2.0-HT, Hamamatsu, Japan) to acquire whole slide images. Quantification of IHC whole slide images was performed using the QuPath software.

### Statistical analysis

Two-tailed Student's t-test was used for all comparisons between two groups, with Welch's correction used for groups with unequal variances. One-way ANOVA was performed for all comparisons involving three or more groups, and two-way ANOVA was done for all growth curve analysis. Fisher's exact test was used for all association testing and Spearman correlation for all gene expression correlation testing. A $p$ value of <0.05 was considered statistically significant for all experiments. All statistical analysis was done using the GraphPad Prism Software Version 9.4.1.

### Reporting summary

Further information on research design is available in the Nature Portfolio Reporting Summary linked to this article.

## Data availability

The METABRIC publicly available data used in this study are available in the cBioportal database: https://www.cbioportal.org/study/summary?id=brca_metabric. The TCGA PanCancer Atlas publicly available data used in this study are available in the cBioportal database: https://www.cbioportal.org/study/summary?id=brca_tcga_pan_can_atlas_2018. The Charafe-Jauffret publicly available cell lines data used in this study are available in the Supplementary Table 2 in: https://aacrjournals.org/cancerres/article/69/4/1302/552886/Breast-Cancer-Cell-Lines-Contain-Functional-Cancer. The remaining data are available within the Article, Supplementary Information or Source Data file. Source data are provided with this paper.

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

## Acknowledgements

This study was supported by funds to M.R. and P.S.O. from the Canadian Cancer Society Challenge Grant (Award #707511), to M.R. through generous funding from Dr. Sami Qureshi and Family, and to W.M. by a Canadian Institute of Health Research Canada Graduate Scholarship Doctoral Award (#170849). This research was funded in part by the Ontario Ministry of Health and Long Term Care.

## Author contributions

Conceptualization, design, and development of methodology: W.Z., W.M., and M.R. Acquisition, analysis, and interpretation of data: W.Z., W.M., K.M., V.S., M.B., P.S.O., and M.R. Material support: C.G. provided the KBP cell line. Writing, review, and revision of the manuscript: W.Z., W.M. and M.R.

## Competing interests

The authors declare no competing interests.
