## [Peer Review File · Nature Communications]

Caspase-1-dependent spatiality in triple-negative breast cancer and response to immunotherapyREVIEWER COMMENTS

Reviewer #1 (Remarks to the Author): with expertise in breast cancer, immunology

This study investigated the role of tumor-intrinsic caspase 1 in shaping the immune microenvironment of TNBC tumors. The topic is of significant interest in the fields of breast cancer therapy and immuno-oncology. The study is well designed and executed, and the data convincingly identify caspase 1 as a promising therapeutic target in TNBC, and, therefore, of significant clinical relevance. One of the strengths of the study is the use of in vitro and in vivo models and human patient samples to ensure reliable findings. Appropriate controls and rescue experiments are also a strength.

Specific comments:

Fig. 2A: Caspase 1 IF staining in patient samples. The caspase 1 expressed is not specific to the tumor cells. It is equally expressed in the stromal and epithelial areas. Is stromal expression expected and what is the mechanism? Please discuss. It would also be helpful to include a control that can validate the specificity of caspase-1 staining. For example, no primary antibody, or using blocking antibody, or alternative caspase-1 specific antibody. Is there a correlation between caspase-1 mRNA and T cell infiltrate in TCGA dataset? The authors have performed similar analysis in 1B.

Reviewer #2 (Remarks to the Author): with expertise in cancer immunology

In the present manuscript, the authors have studied the mechanism underlying the limited efficacy of immune-check blockade in TNBC. They identify estrogen receptor alpha as a natural regulator of caspase 1 and demonstrated that its absence promotes unattenuated expression of caspase 1 and secretion of IL-1b. In turn, this shapes the tumor microenvironment, by promoting infiltration of pro-tumoral macrophages and reduced CD8+ T cells. Regulating caspase 1 activity sensitizes TNBC to ICB therapy.

Overall, this is a very interesting work, well written and well executed. To my opinion, however, several issues should be addressed before it may be considered for publication:

Major concerns:

1. The capacity of caspase 1 inhibitors to sensitize TNBC to ICB should be demonstrated in at least one more TNBC model.
2. IHC staining of IL-1b, and/or IL-18 should be included to demonstrate that it is secreted from TNBC tumor cells.
3. The levels of cleaved caspase 1 and production of IL-1b should be thoroughly characterized in the KBP tumor model.
4. The reduction in caspase 1 transcription levels upon addition of ERa should be shown in additional human cell lines (at least one more).
5. Do tumors treated with PD1 in caspase 1 KO eventually escape?
Can the authors speculate what the mechanism is? Along these lines, it seems like the effect of caspase 1 inhibitor is stronger. Is that true? It is highly advised to include survival plots for this experiment.

Minor concerns:

1. For most figures, the titles should probably be revised to better describe the key findings. for example,
2. In figures 3A-3E and 4B, how come the percentages of F4/80 are higher in the total cells compared to their calculated percentages from the CD45+ cells?
3. Representative FACS plots, demonstrating the gating strategy of each cell type, should be included.
4. The illustrations in Figures 1F, 2E, and 4A should be improved.
5. In Figure 5a, the site of tumor inoculation should be altered to indicate the mammary fat pad.

Reviewer #3 (Remarks to the Author): with expertise in breast cancer, immunology

This study aims to investigate the potential of caspase-1 inhibition in reversing ICB resistance in triple-negative breast cancer therapy. However, the author only presents preliminary experimental results in this manuscript, which may not fully meet the publication criteria.

1. Although the author utilized an IHC experiment to illustrate the association between

caspase-1 and spatial immunophenotype in human TNBC, further experiments are required to validate this hypothesis.

2.The author employed murine TNBC model to assess the therapeutic efficacy of caspase-1 inhibition, necessitating a humanized tumor model to demonstrate its ability in inducing an immunoreactive TIME within human tumors.

3.This study demonstrates that IL1 β mediated macrophage recruitment in TNBC. However, the author only measured the macrophage infiltration without detecting their polarization phenotype, which is crucial for TIME remodeling.

4.Despite conducting many allograft-bearing mice experiments throughout this study, no specific therapeutic regimen was proposed by the author at its conclusion.

Reviewer 1

1. Fig. 2A: Caspase 1 IF staining in patient samples. The caspase 1 expressed is not specific to the tumor cells. It is equally expressed in the stromal and epithelial areas Is stromal expression expected and what is the mechanism? Please discuss.

This is an important point and requires clarification. Caspase-1 is expressed in both the epithelial and stromal compartments. Caspase-1 expression is well-documented and expected from immune cell types including B cells, neutrophils, and macrophages. As we demonstrate, immune cells are abundant in the TNBC stroma and correlate with caspase-1 expression in that compartment. Respectfully, a detailed analysis of the mechanism of caspase-1 expression in immune/stromal cells is beyond the scope of this paper. We have added the sentence “Stromal expression of caspase-1, expected from immune cell infiltrates, was not associated with macrophage subtype (S4k - m).” (lines 128-9). The new, accompanying data (S4k - m) provides additional evidence that epithelial, and not stromal caspase-1 coordinates the class switch from M1-like to alternatively activated M2-like TAMs.

2. It would also be helpful to include a control that can validate the specificity of caspase-1 staining. For example, no primary antibody, or using blocking antibody, or alternative caspase-1 specific antibody.

We have added controls: (1) a no primary antibody control and (2) staining an alternative tissue, human palatine tonsil, as a positive control to validate the specificity of the anti-caspase-1 antibody (Fig S4a and b).

3. Is there a correlation between caspase-1 mRNA and T cell infiltrate in TCGA dataset? The authors have performed similar analysis in 1B.

As suggested, we reanalyzed TCGA (and added Fig. S4f and g). Not surprising, we found a positive correlation between caspase-1 mRNA and T cell lymphocyte infiltrate, confirming our previous finding that TNBC (caspase-1 high) have more abundant immune infiltrates (which can also express caspase-1 – see reply to Reviewer 1, point 1). To discuss these findings, we have added “Accordingly, a positive correlation was identified between caspase-1 mRNA and T cell lymphocyte infiltration in the TCGA PanCancer Atlas dataset (Fig. S4f and g)” (lines 124-5). These findings are consistent with our hypothesis that intratumoral caspase 1 determines the location and activity of T cells within the tumor (Fig. 2), not the number of T-cells.

Reviewer 2

The capacity of caspase 1 inhibitors to sensitize TNBC to ICB should be demonstrated in at least one more TNBC model.

We agree that it is important to prove that our findings are generalizable. New results using a 4T1 TNBC Mouse allograft model are shown together with the sentence “and in a 4T1 murine TNBC allograft model (Fig. S8b - f)”. (lines 181-2) Also, as per Reviewer 3 (see point 2, below) we repeated the experiments in a humanized tumor model to demonstrate the therapeutic efficacy of caspase-1 inhibition in inducing an immunoreactive TIME within human tumors.

2. IHC staining of IL-1b, and/or IL-18 should be included to demonstrate that it is secreted from TNBC tumor cells

Thank you for this suggestion. We confirmed that IL1 β is elevated in TNBC compared to ER positive tumors and this data now appears as a new figure (Fig. S4c). As expected, we find that IL1 β is present in both stroma (likely from immune cells) and epithelium (from tumor). However, because these human tissues blocks are a limited resource, we could not verify these findings using other methods. We respectfully remind the reviewer of the experiments shown in Fig. S1f and g which demonstrate secretion of active IL1 β (and IL18) from triple-negative MDA MB231 cells in an ER α - and caspase 1-dependent fashion.

3 .The levels of cleaved caspase 1 and production of IL-1b should be thoroughly characterized in the KBP tumor model.

We agree that the initial submission contained no description of the KBP model, specifically of IL1 β production in these cells and its dependence on caspase 1 processing. Therefore, we have reorganized Fig. S5 and have included a thorough description of the findings. We show that similar to human TNBC cells, in KBP cells pro-caspase-1 production is ER α -, and ETS1-dependent and the production of enzymatically active caspase 1 is ER α -, ETS1-, and VX-765-dependent Fig. S5a – c). Furthermore, mirroring findings in human TNBC, IL1 β gene expression is not caspase-1 dependent (Fig. S5e). Additionally, the production of secreted IL1 β is 100X greater in KBP cells derived from KBP tumor compared to KBP cells grown in monolayer, consistent with our previous finding (Shen et al., 2017, ref 9) that IL1 β expression depends upon tumor cell-stromal interaction. We show here that tumoral IL1 β secretion is caspase-1-dependent. We have added: “KBP cells were reengineered with modifications in the IL1 β processing pathway according to our findings in human TNBC cells (Fig S5a – d). As expected, caspase-1 did not affect IL1 β gene expression and because of the absence of paracrine stromal-tumor cell interaction, KBP cells grown in monolayer produced minimal IL1 β (9) (Fig S5 e and f). However, KBP cells isolated from KBP tumors produced 100-fold more IL1 β , and this was caspase-1 dependent. Given that KBP cells faithfully reproduced the findings in human

TNBC cells, genetically modified KBP cells were orthotopically injected into the mammary fat pads of FVB/NJ mice to create mammary tumors.” (lines 142-8)

4. The reduction in caspase 1 transcription levels upon addition of ER α should be shown in additional human cell lines (at least one more).

We agree and have added data for the TNBC cell line HCC1143 and have updated the manuscript accordingly: “Conversely, ER α overexpression resulted in pro-caspase-1 downregulation in TNBC cells and a decrease in secreted IL1 β and IL18 in conditioned media (Fig. S1f) and, confirming the presence of functional caspase-1, a decrease in the ability of the media to promote macrophage migration (Fig. S1g) (lines 92-5).

5. Do tumors treated with PD1 in caspase 1 KO eventually escape? Can the authors speculate what the mechanism is? Along these lines, it seems like the effect of caspase 1 inhibitor is stronger. Is that true? It is highly advised to include survival plots for this experiment.

This is an important question and so we repeated the experiment shown in Fig 4a, performing K-M survival plots. This new data appears in the manuscript as Fig S8a. The plots show that mice bearing caspase-1 KO allografts and treated with anti-PD1 demonstrate the longest median survival, almost twice that of mice bearing wild type tumors. We have added the sentence “growth, weight, and median survival” (line 177). It is the case, as the reviewer points out, that the effect of caspase-1 KO is stronger than anti-PD1. This is likely because the reduction in protumoral M2-like TAMs not only reboots anti-PD1 sensitivity, but also because caspase-1 neutralization eliminates other protumoral effects of IL1 β such recruitment of other protumoral inflammatory cells, tumor growth, invasion, angiogenesis, metastases, stemness and epithelial-mesenchymal transition, as discussed in the manuscript.

As the reviewer suggests caspase-1 KO tumors treated with anti-PD1 eventually escape. We speculate that escape is due to genetic instability in tumor cells that allows outgrowth of therapy resistant clones. Additionally, tumor cells may undergo further immune escape through the loss of HLA or through other mechanisms that result in the loss of antigenicity/immunogenicity. To reflect this, we have added new text to the discussion: “The addition of cytotoxic chemotherapy could also reduce tumor escape noted in our mouse experiments, where immune modulation alone, was undertaken.” (lines 259-60).

Minor

concerns:

1. For most figures, the titles should probably be revised to better describe the key findings.

We clarified the figure legend titles and changes appear in red in the marked version of the manuscript.

2. In figures 3A-3E and 4B, how come the percentages of F4/80 are higher in the total cells compared to their calculated percentages from the CD45+ cells?

We believe that there are two reasons for this finding: 1) in the CD45+ population, macrophages were identified as double (F4/80+ and CD11b+) stained cells, whereas in the total cell population macrophages were identified as F4/80+ single-stained cells. Single-stained cells will be more numerous than double-stained, which would increase the fraction of those cells and, 2) the method to quantify the proportion of macrophages was different in the CD45+ and total cell populations. In the CD45+ we used flow cytometry, whereas IHC staining was used to quantify macrophages in the total cell population. Antibody affinity could differ between these two methodologies making direct comparisons misleading.

3. Representative FACS plots, demonstrating the gating strategy of each cell type, should be included.

Please see Fig. S6b where we added representative FACs plots for the data shown in Fig. S6a.

4. The illustrations in Figures 1F, 2E, and 4A should be improved.

As recommended, we remade these figures.

5. In Figure 5a, the site of tumor inoculation should be altered to indicate the mammary fat pad.

Thank you for pointing this out. The new figure was changed to demonstrate that these are orthotopic tumor injections.

Reviewer 3

1. Although the author utilized an IHC experiment to illustrate the association between caspase-1 and spatial immunophenotype in human TNBC, further experiments are required to validate this hypothesis.

We appreciate this feedback. In addition to multiplex IHC, there are additional methodologies that were considered to characterize the spatial immunophenotype of TNBC including laser capture microdissection and gene expression profiling, scRNA-seq of flow-selected stromal and immune populations, imaging time-of-flight mass spectrometry, spatial transcriptomics etc. However, since these prospectively collected and annotated formalin-fixed paraffin-embedded

sections were a limited institutional resource, we were limited to multiplex IHC. We respectively suggest that an in-depth analysis of human cancers is beyond the scope of this paper. In fact, we are currently collecting T1a/b TNBCs in a phase 1/2 RCT for this purpose, the results to be reported in a follow up study. The objective of the current manuscript was to identify the mechanism of caspase-1 expression in TNBC, to verify the mechanism in mouse models (KBP, 4T1, MDA MB231 in humanized mice), and from a therapeutic perspective, to demonstrate that anti-caspase-1 treatment of TNBC exposes a vulnerability to anti-PD1. The human data displayed in Fig. 2 expands on our findings in Shen et al., 2017 (ref 9) showing that compared to luminal subtypes, TNBCs can be characterized by their expression of components of the inflammasome, including caspase-1. The novel finding in Fig. 2 is that caspase-1 coordinates a class switch from classically activated (M1-like) to alternatively activated (M2-like) TAMs. The reviewer requests further experiments to illustrate that caspase-1 and the immunophenotype are associated, and we provide these in new data from mouse models (Fig 4d and h Fig. S8e and f) where a similar caspase-1-dependent TAM class switch is verified.

2. The author employed murine TNBC model to assess the therapeutic efficacy of caspase-1 inhibition, necessitating a humanized tumor model to demonstrate its ability in inducing an immunoreactive TIME within human tumors.

This is an important suggestion and so we have completed additional experiments in humanized mice which are now described in the manuscript as follows: “Demonstrating the ability of VX-765 to induce an immunoreactive TIME within human tumors, treatment of humanized mice bearing MDA-MB231 xenografts decreased CD14+ HLA-DR+ macrophages and increased activated granzyme B positive (GrB+) CD8+ T cells, with a trend ($p = 0.1$) towards decreased tumor size, (Fig. S6g) (lines 154-7).

3. This study demonstrates that IL1 β mediated macrophage recruitment in TNBC. However, the author only measured the macrophage infiltration without detecting their polarization phenotype, which is crucial for TIME remodeling.

Thank you for identifying this oversight. We had these data but including them was overlooked when we prepared the original submission. The data which appear in the new figures Fig. 4d and h (KBP model) and Fig. S8 (4T1 model) reflect what we see in the human tumors (Fig 2), which is an association between caspase-1 expression in tumor cells and the presence of alternatively activated M2-like macrophages. The findings are described as follows: “Caspase-1 KO was associated with reduced F4/80+ macrophages and reflecting our findings in human TNBC, a class switch from alternatively activated (CD206+, M2-like) to classically activated (CD206-, M1-like) TAMs.” (lines 177-9).

4.Despite conducting many allograft-bearing mice experiments throughout this study, no specific therapeutic regimen was proposed by the author at its conclusion.

We agree and have added to the discussion “In the meantime alternative compounds such as canakinumab, or the highly selective recombinant IL1 β antagonist anakinra (45) should be considered for combination therapy with ICB. These compounds could be tested within the setting of a clinical trial in patients with Stage II/III TNBC receiving standard-of-care pembrolizumab ICB/taxane-platinum/anthracycline-based chemotherapy. The addition of cytotoxic chemotherapy could also reduce tumor escape noted in our mouse experiments, where immune modulation alone, was undertaken.” (lines 255-60).

REVIEWERS' COMMENTS

Reviewer #1 (Remarks to the Author):

The authors adequately addresses the issues brought up in the first round of review.

Reviewer #2 (Remarks to the Author):

The authors addressed all of my concerns and I have no further comments.

Reviewer #3 (Remarks to the Author):

The authors have performed additional experiments to address most of the concerns from the reviewers. I do not have any more comments.